

# A new ophiacanthid brittle star (Echinodermata, Ophiuroidea) from sublittoral crinoid and seagrass communities of late Maastrichtian age in the southeast Netherlands

Ben Thuy[1], Lea Numberger-Thuy[1] and John W.M. Jagt[2]

[1] Department of Palaeontology, Natural History Museum Luxembourg, Luxembourg City, Luxembourg
[2] Natuurhistorisch Museum Maastricht, Maastricht, The Netherlands

## ABSTRACT

A new species of brittle star, *Ophiomitrella floorae*, is recorded from the lower two meters of the Gronsveld Member (Maastricht Formation), of late Maastrichtian age (c. 66.7 Ma). These relatively fine-grained biocalcarenites reflect shallow-water deposition in a sheltered setting with a relatively firm sea floor and clear waters, under middle sublittoral and subtropical conditions. Associated echinoderm taxa comprise more robust, sturdy-plated ophiomusaid and ophiodermatid brittle stars and numerous bourgueticrinine sea lilies. The new brittle star described herein belongs to a family whose present-day members are predominantly restricted to bathyal depths. Its small size and the exceptional preservation of a single articulated specimen partially wrapped around the stalk of a bourgueticrinine suggest that *O. floorae* n. sp. was probably epizoic and specifically associated with stalked crinoids.

## INTRODUCTION

During recent decades, there has been a renewed interest in macrofossil assemblages from Upper Cretaceous (Campanian–Maastrichtian) strata in the type area of the Maastrichtian Stage (southeast Netherlands, northeast Belgium; *Felder, 1975a*; *Felder, 1975b*). This has resulted in the recovery of numerous previously unrecorded taxa, particularly amongst echinoderms. The former ENCI-Heidelberg Cement Group quarry at Sint-Pietersberg, south of the city of Maastricht (Figs. 1 and 2), is a key locality in the area. Here the lower/middle portion of the Maastricht Formation (Valkenburg, Gronsveld, Schiepersberg and Emael members) has yielded a range of brittlestar taxa over recent years, amongst which sturdy-plated ophiomusaids and ophiodermatids predominate (*Jagt, 1999c*; *Jagt, 1999d*; *Jagt, 2000a*). Smaller-sized species are much rarer and often occur as dissociated ossicles of disc and arms only. An articulated specimen of an ophiacanthid partially wrapped around the stalk of a bourgueticrinine crinoid from the lower Gronsveld Member (*Jagt, 2000a*), in which obrution-related echinoderm Lagerstätten have been recorded between the St.

Corresponding author
Ben Thuy, bthuy@mnhn.lu

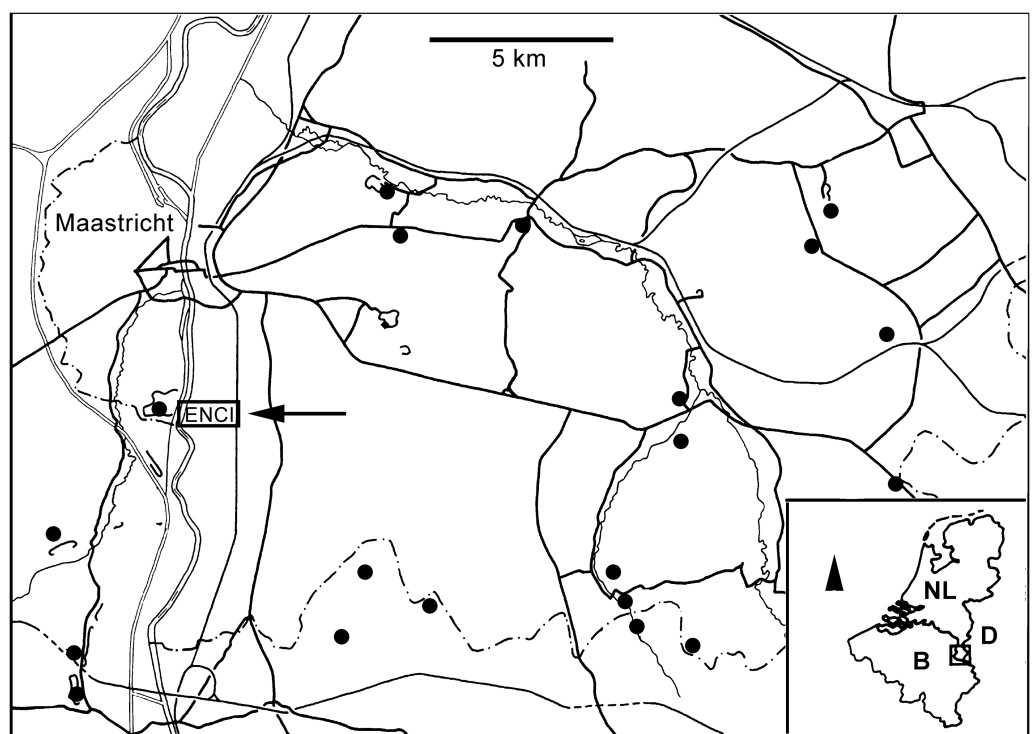

**Figure 1 Locality map.** Map of southern Limburg and contiguous areas in Belgium and Germany, representing the extended type area of the MaaSt.richtian St.age and showing the location of the former ENCI-HeidelbergCement Group quarry (modified from *Jagt et al., 2020*).

Pieter and ENCI horizons (Fig. 3) (see *Jagt et al., 1998*; *Jagt, 1999b*), provided the impetus for the present note. In fact, it represents one of the very few known examples of an articulated ophiacanthid fossil. Albeit illustrated and described in previous studies, it has not been adequately assigned. Thanks to recent advances in ophiuroid phylogeny and in the systematic assessment of dissociated skeletal plates (in particular lateral arm plates) and their microstructures, it was possible to carry out a more detailed morphological analysis and attempt a conclusive taxonomic interpretation of the material, thus filling a critical gap in the fossil record of ophiacanthid brittle stars around the time of their demise at shallow depths (*Thuy, 2013*).

## Stratigraphical setting

The lower portion of the Maastricht Formation at the former ENCI-Heidelberg Cement Group quarry comprises comparatively fine-grained, poorly indurated, pale yellow biocalcarenites with a diverse macrofossil content, in particular in the Valkenburg and Gronsveld members (Fig. 3). On the basis of recent cyclostratigraphical and chronostratigraphical age models for the type Maastrichtian (*Keutgen, 2018*), the base of the Valkenburg Member (i.e., the contact between the Gulpen and Maastricht formations or Lichtenberg Horizon) can be dated at 66.8 Ma, and the base of the overlying Gronsveld Member (St Pieter Horizon) at 66.7 Ma. The latter horizon is thought to represent the early

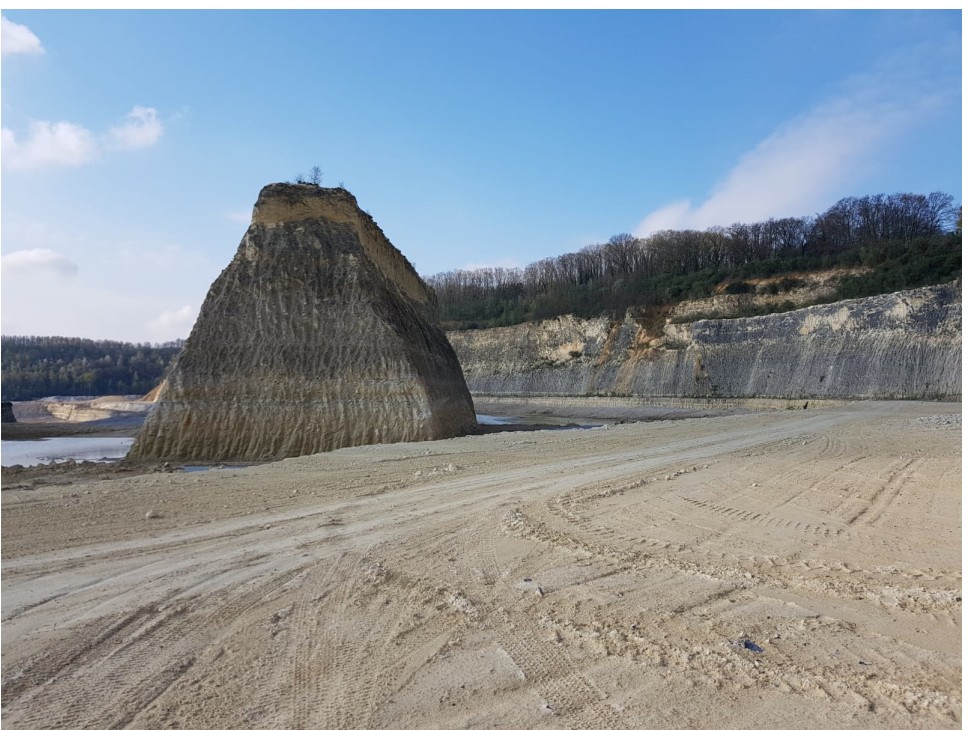

**Figure 2  ENCI-HeidelbergCement Group quarry.** The northeaSt. corner of the former ENCI-HeidelbergCement Group quarry (MaaSt.richt), looking southweSt. (Spring 2019); the level with tyre tracks corresponds roughly to the lower Gronsveld Member (St. Pieter and ENCI horizons; see Fig. 3; photograph by Mart J.M. Deckers).

stages of a transgression from a relative lowstand during a tectonic inversion phase, while the overlying Gronsveld Member represents a relative highstand during tectonic relaxation, with the maximum flooding surface situated around the middle of this unit (*Schiøler et al., 1997*).

In more general terms, referring to the area west of the River Maas (Meuse), the lowest unit of the Maastricht Formation, the Valkenburg Member, comprises poorly indurated, white-yellowish to yellowish-grey, fine- to coarse-grained biocalcarenites, with greyish brown flint nodules of varying sizes. The overlying Gronsveld Member consists of poorly indurated, white-yellowish to yellowish-grey, fine- to coarse-grained biocalcarenites, with small, light to dark greyish-brown flint nodules of varying sizes and shapes occurring in the lower part. In the higher portion they are arranged in more or less regular beds of light-grey to greyish blue nodules (Fig. 3).

The lower portion of the Maastricht Formation has been considered to represent a gravelly intrabiomicrosparite, with regional currents constant enough for horizontal displacement of sediment particles over the entire platform, at sublittoral depths, judging from benthic foraminifera, ostracods and numerous plant fossils, including terrestrial forms such as conifers, and sheltered from oceanic influence (*Villain, 1977*; *Jagt, 1999a*; *Jagt & Jagt-Yazykova, 2012*). Frequent sediment reworking resulted in homogenisation

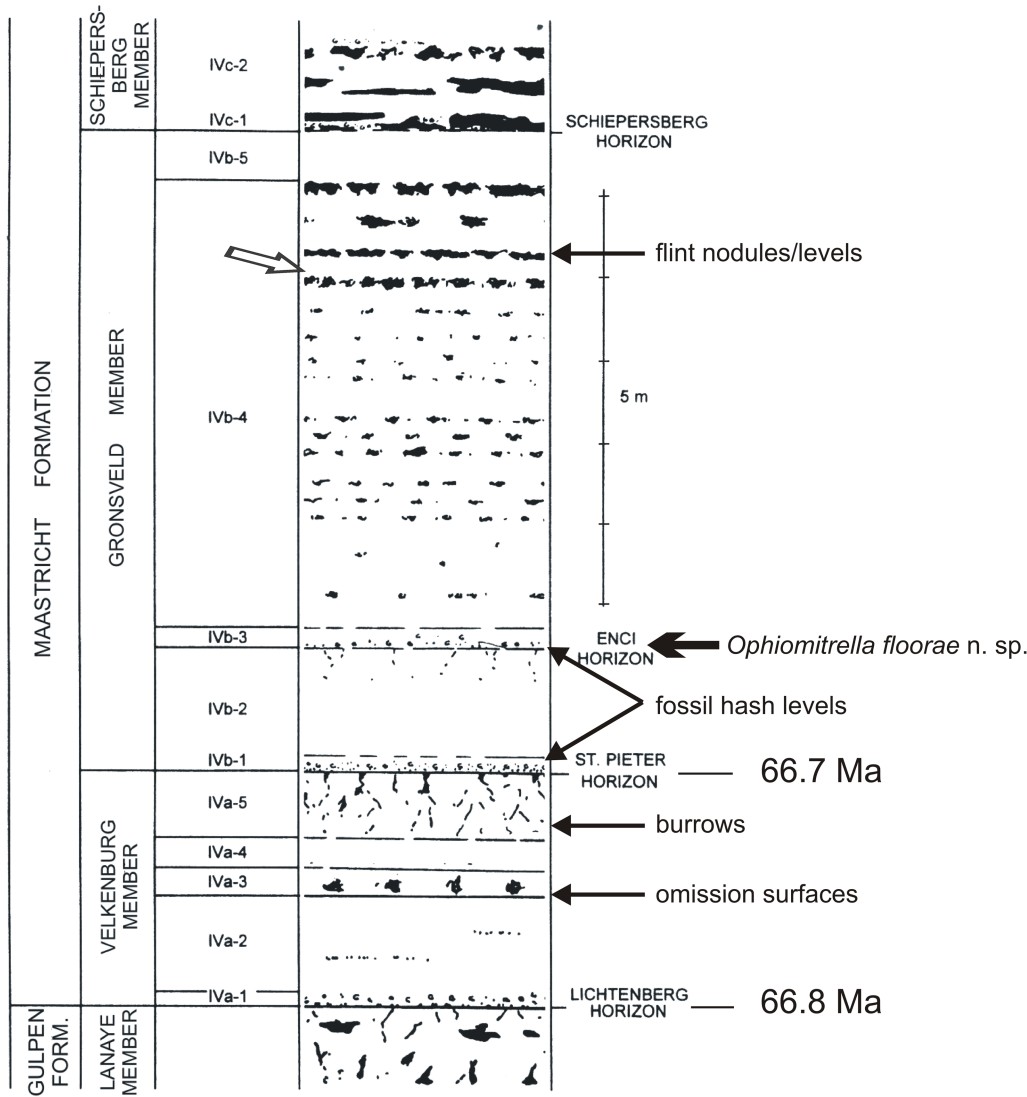

**Figure 3** **Litholog of the lower MaaSt.richt Formation.** Log (modified from *Felder & Bosch, 1998*), with the St. Pieter and ENCI horizons in the lower part of the Gronsveld Member. The arrow in the higher part of that unit refers to one of the more spectacular St.orm levels in the section (for details, see *Jagt et al., 2019*).

over depths of a few decimetres, leading to a relatively firm sea floor and clear waters. This setting has been interpreted as middle sublittoral, under subtropical conditions and with sea grass communities (*Liebau, 1978*; *Jagt et al., 2019*).

On evidence of index forms amongst coleoid cephalopods (*Christensen, Schmid & Schulz, 2005*; *Jagt & Jagt-Yazykova, 2019*) and inoceramid bivalves (*Jagt & Jagt-Yazykova, 2018*), the lower portion of the Maastricht Formation has been shown to be of late, though not latest, Maastrichtian age, thus corroborating age assignments on the basis of dinoflagellates and calcareous nannoplankton (see *Schiøler et al., 1997*; *Keutgen, 2018*). All these biota allow correlation of these shallow-water biocalcarenites along the fringes of Palaeozoic

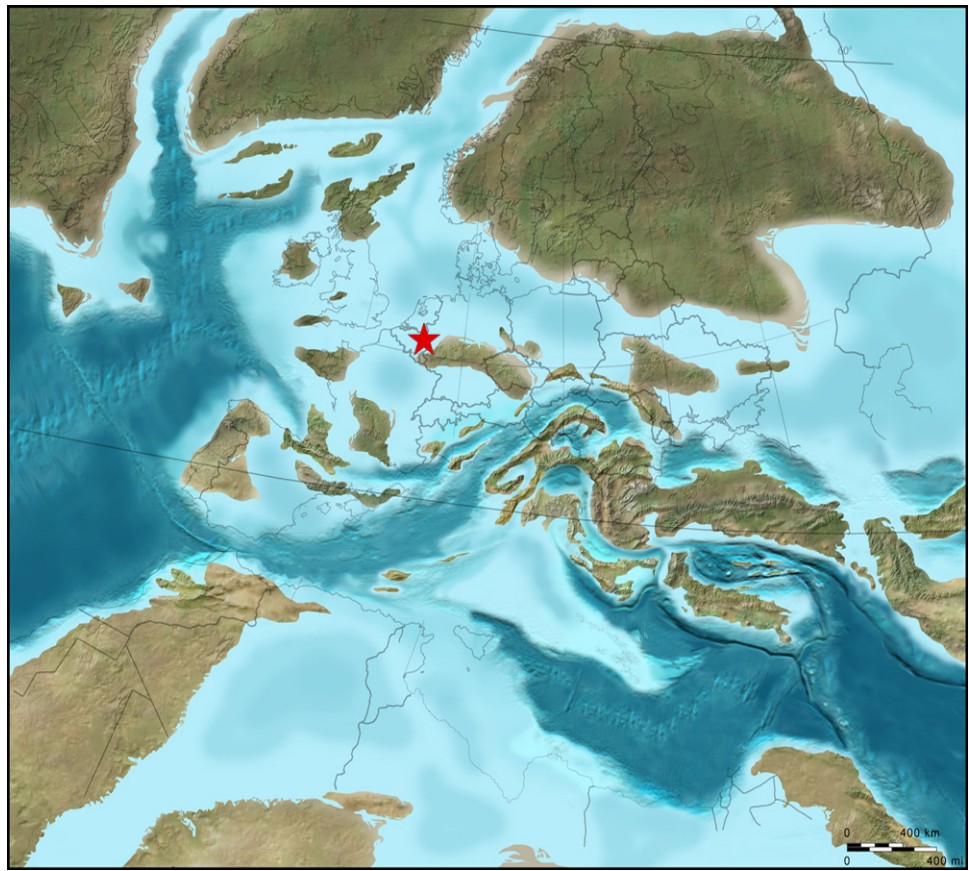

**Figure 4  Plaeogeographical map.** ReconSt.ruction of oceans, epicontinental seas and land masses during the late Late Cretaceous (c. 75 Ma; ©2012 Colorado Plateau GeosySt.ems Inc.).

massifs (Fig. 4) with coeval levels in deeper-water settings (white chalk, Schreibkreide) elsewhere in Europe (northern Germany, Denmark and Poland).

## Previous work on ophiuroids

Earlier records of late Maastrichtian echinoderms in the type area of the Maastrichtian Stage have demonstrated several Lagerstätten, comprising mostly bourgueticrinine crinoids (and other comatulids associated; see *Jagt et al., 1998*; *Jagt, 1999b*) as well as lesser numbers of echinoids (*Jagt, 2000b*), asteroids (*Jagt, 2000c*; *Blake & Jagt, 2005*; A.S. Gale & J.W.M. Jagt, work in progress) and ophiuroids.

Brittle stars from the lower Gronsveld Member (*Jagt, 1999c*; *Jagt, 1999d*; *Jagt, 2000a*) include mostly semi-articulated individuals of the sturdy-plated *Ophiomusium granulosum* (*Roemer, 1840–1841*) (= *Ophiura* (*Aspidura*) *subcylindrica Von Hagenow, 1840*), *Ophiotitanos serrata* (*Roemer, 1840–1841*) (= *Ophiura parvisentis Spencer, 1905–1908*; *Ophioglypha gracilis Valette, 1915*) and *Ophiopeza*? *hagenowi* (*Wienberg Rasmussen, 1950*) (see *Wienberg Rasmussen, 1950*; *Jagt, 2000a*; *Ishida et al., 2018*). Other taxa, such as *Trichaster*? *ornatus* (*Wienberg Rasmussen, 1950*) and *Ophiothrix*? *bongaertsi Kutscher &*

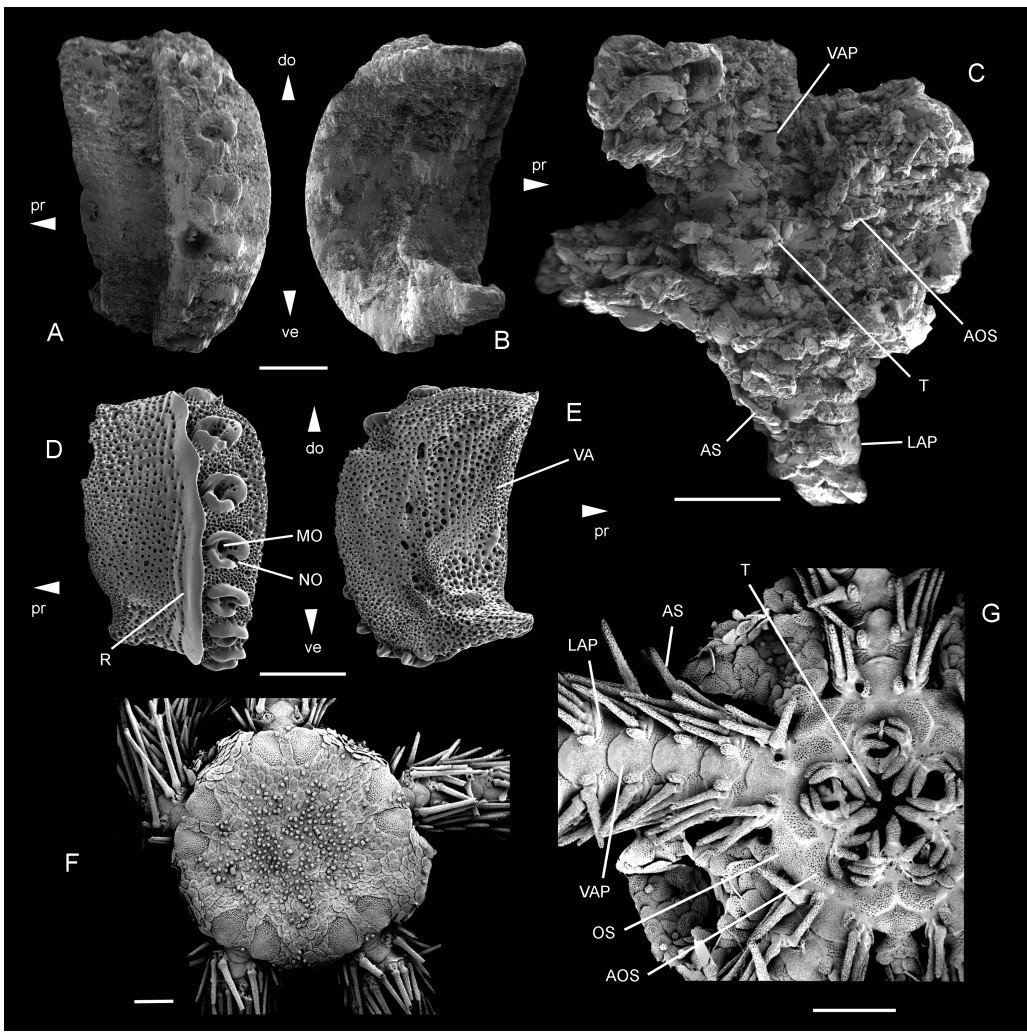

**Figure 5** *Ophiomitrella. floorae* n. sp., from the lower Gronsveld Member (MaaSt.richt Formation; St. Pieter and ENCI horizons) at the former ENCI-HeidelbergCement Group quarry, MaaSt.richt, the Netherlands. A–B: holotype (NHMM JJ 5104), dissociated proximal lateral arm plate in external (A) and internal (B) views; C: paratype (NHMM K 3387), articulated disc with basal arm segments in ventral view. *Ophiomitrella granulosa* Recent, as a close living relative of *O. floorae* n. sp. D–E: proximal lateral arm plate (SMNH-121224) in external (D) and internal (E) views; F–G: complete individual (MNHNEcOs22429) in dorsal view (F) and with detail of ventral disc skeleton (G). Abbreviations: AOS, adoral shield; AS, arm spine; do: dorsal; LAP, lateral arm plate; MO, muscle opening; NO, nerve opening; OS, oral shield; pr, proximal; R, ridge; T, tooth; VA, vertebral articulation; VAP, ventral arm plate. Scale bars equal 0,25 mm in A–B and D–E, and 1 mm in C and F–G. Scanning electron microscope images of Figs. F-G by Sabine St.öhr.

*Jagt, 2000*, are much rarer and occur only as dissociated vertebrae and lateral arm plates, respectively (*Jagt, 2000a*).

## MATERIALS & METHODS

*Jagt (2000a)*, *Jagt (2000b)* and *Jagt (2000c)* records of the ophiacanthid *Ophiacantha*? *danica Wienberg Rasmussen, 1952* from the lower Gronsveld Member was based on a single, articulated individual in life position around a crinoid stalk (NHMM K 3387), as well as a single isolated lateral arm plate (NHMM JJ 5104) obtained from the sieving residues of a bulk sediment sample from the same deposits. It is this material which is revised herein. The material described herein was illustrated and/or mentioned in previous studies (*Jagt, 2000a*; *Thuy, 2013*). For the purpose of the present reassessment, dissociated lateral arm plates and the disc of the articulated specimen, detached from the matrix, were mounted on a stub and gold-coated for scanning electron microscopy (LEO 1530 Gemini). Morphological terminologies follow *Stöhr, O'Hara & Thuy (2012)* for general skeletal features, *Thuy & Stöhr (2011)* for lateral arm plates and *Hendler (2018)* for the mouth skeleton. We adopt the classification proposed by *O'Hara et al. (2017)* and *O'Hara et al. (2018)*. To denote the repositories of the material described and illustrated here, the following abbreviations are used: NHMM, Natuurhistorisch Museum Maastricht, Maastricht, the Netherlands; SMNH, Swedish Museum of Natural History, Stockholm, Sweden; MNHN, Muséum national d'Histoire naturelle, Paris, France.

The electronic version of this article in Portable Document Format (PDF) will represent a published work according to the International Commission on Zoological Nomenclature (ICZN), and hence the new names contained in the electronic version are effectively published under that Code from the electronic edition alone. This published work and the nomenclatural acts it contains have been registered in ZooBank, the online registration system for the ICZN. The ZooBank LSIDs (Life Science Identifiers) can be resolved and the associated information viewed through any standard web browser by appending the LSID to the prefix http://zoobank.org/. The LSID for this publication is: urn:lsid:zoobank.org:pub:9BE69BFD-69FE-4671-BC94-0DF402806A75. The online version of this work is archived and available from the following digital repositories: PeerJ, PubMed Central and CLOCKSS.

## RESULTS

### Systematic palaeontology

Class Ophiuroidea *Gray (1840)*
Subclass Myophiuroidea *Matsumoto (1915)*
Infraclass Metophiurida *Matsumoto (1913)* (crown-group of Ophiuroidea)
Superorder Ophintegrida *O'Hara et al. (2017)*
Order Ophiacanthida *O'Hara et al. (2017)*
Suborder Ophiacanthina *O'Hara et al. (2017)*
Family Ophiacanthidae *Ljungman (1867)*
Genus *Ophiomitrella Verrill (1899)*
*Ophiomitrella floorae* n. sp.
Figs. 5A–5C

Etymology: Named after Floor Jansen, lead singer of the Finnish band Nightwish in recognition of her long-standing career in metal, her general interest in all things (palaeo) biological and her and the band's use of fossils for artwork (*Metal Mike, 2020*).

Holotype: NHMM JJ 5104

Type locality and stratum: lower Gronsveld Member (Maastricht Formation; St. Pieter and ENCI horizons) at the former ENCI-Heidelberg Cement Group quarry, Maastricht, the Netherlands.

Paratype: NHMM K 3387

Note that most of the specimens cited as *Ophiacantha*? *danica* in *Jagt (2000a)* have subsequently been re-assigned to *Ophiogaleus danicus* by *Thuy (2013)* and *Ophiotreta striata* (*Kutscher & Jagt, 2000*). Of the remaining specimens, only the articulated disc of *Jagt (2000a)*, *Jagt (2000b)* and *Jagt (2000c)* plate 2, figure 7 (NHMM K 3387) and the lateral arm plate of *Thuy (2013)* figure 32-2 (NHMM JJ 5104) are unambiguously assignable to the species described herein. All other specimens either are too poorly preserved for a conclusive assessment or belong to a different species.

Diagnosis: Small species of *Ophiomitrella* with high lateral arm plates showing up to eight large spine articulations and a very weak and fine vertical striation; large, wide adoral shields; two to three large, conical oral papillae sensu lato and a single large, conical ventralmost tooth.

Description of holotype: NHMM JJ 5104 (Figs. 5A–5B) is a dissociated proximal lateral arm plate, almost two times taller than long; dorsal edge concave due to a strong constriction; distal edge strongly and regularly convex; proximal edge weakly concave and devoid of spurs; ventral portion of lateral arm plate not protruding. Outer surface with finely meshed stereom and a very weak, fine vertical striation close to ridge of spine articulations. Eight large, ear-shaped spine articulations on a strongly elevated distal portion of lateral arm plate; row of spine articulations proximally bordered by thick, conspicuous, straight ridge; spine articulations each consisting of large muscle opening enclosed by dorsal and ventral lobes forming round, continuous ring, and separated from smaller nerve opening by well-developed sigmoidal fold; weak dorsalward increase in size of spine articulations and distance between them. Ventral edge of lateral arm plate oblique; tentacle notch invisible in external view; row of spine articulations protruding ventralwards. Inner side of lateral arm plate with large, well-defined vertebral articulation shaped like slightly rotated digit one with an expanded nose; tentacle notch small but well defined, distally bordering thickened ventral edge of lateral arm plate; poorly defined vertical furrow running parallel to row of spine articulations but presence of perforations ambiguous due to insufficient preservation.

The paratype (NHMM K 3387, Fig. 5C) is an articulated skeleton with an arm wrapped around a bourgueticrinine stalk; the proximal arm portions show lateral arm plates similar to the holotype; the disc is poorly preserved due to coarse recrystallisation, blurring all details on the dorsal side; ventral side of the disc preserving a few details of the skeleton; four arm bases preserved intact, showing strongly recrystallised lateral and ventral arm plates and ventral arm spines; lateral arm plates identical to holotype; ventral arm plates with a strongly convex distal edge, deeply incised lateral edges and a pointed proximal tip;
arm spines at least as long as one arm segment; adoral shields large and wide; two to three large, conical oral papillae sensu lato and a large, conical ventralmost tooth.

## DISCUSSION

The material described herein unambiguously belongs to the family Ophiacanthidae as defined by *O'Hara et al. (2018)* on account of the large, ear-shaped spine articulations proximally bordered by a sharply defined ridge, the non-protruding ventral portion of the lateral arm plates, and the shape of the ridge on the inner side of the lateral arm plates. Within this family, several clades have been resolved using molecular evidence (*O'Hara et al., 2017*), but only very few agree with previously defined genera (e.g., *Ophioplinthaca*). Most traditional ophiacanthid genera are poly- or paraphyletic, challenging the diagnostic value of the characters used to define these taxa (*O'Hara et al., 2017*). In contrast, patterns in lateral arm plate morphology seem to agree with molecular evidence in many aspects (*O'Hara et al., 2014*; *Thuy & Stöhr, 2016*), corroborating that lateral arm plates can be used to constrain the position of a species within the family Ophiacanthidae (*Thuy, 2013*).

In the light of this conclusion, and due to the poor preservation of the single articulated individual, we have chosen the dissociated proximal lateral arm plate as the holotype of the new species. Microstructural details of the lateral arm plate morphology put the specimens closest to the ophiacanthid genera *Ophiomitrella* and *Ophiacantha*. The mouth skeleton of the paratype specimen corroborates this position. The development of the vertical striation on the outer surface ornamentation of the lateral arm plates, the shape of the ridge proximally bordering the row of spine articulations, and the shape of the vertebral articulation on the inner side of the lateral arm plate excludes all other genera assigned to the family Ophiacanthidae as defined by *O'Hara et al. (2018)*. Both *Ophiomitrella* and *Ophiacantha*, as currently understood, are polyphyletic and require revision (*O'Hara et al., 2018*), hampering a clear-cut genus-level assignment. The type species of *Ophiacantha*, *O. bidentata* (*Bruzelius, 1805*), differs from the specimens described herein in having a more expanded vertical striation on the outer surface and a row of spine articulations that does not protrude ventralwards. We therefore preclude assignment to *Ophiacantha* and favour *Ophiomitrella* instead, although it must be stressed that the type species of the genus, *Ophiomitrella laevipellis* (*Lyman, 1883*), has not been genetically sequenced nor morphologically dissected as yet because it is known only from very valuable historical samples, rarely represented in museum collections.

Within the three clades resolved by molecular evidence and containing the recent species of *Ophiomitrella* (*O'Hara et al., 2017*), the specimens described herein show closest similarities to the species of the clade that contains *Ophiomitrella granulosa* (*Lyman, 1878*) (Figs. 5D–5G) and *O. mensa* (*O'Hara & Stöhr, 2006*). In members of the second clade, in particular *Ophiomitrella conferta* (*Koehler, 1922*) and *O. clavigera* (*Ljungman, 1865*), the vertebral articulation looks more like a digit 1 and lacks the expanded nose. Comparison with members of the third clade of *Ophiomitrella* species, e.g., *O. stellifera* (*Matsumoto, 1917*), was hampered by the lack of information on the microstructures of the lateral arm plate. Thus, as long as the systematic position of the type species is unresolved, assignment

to *Ophiomitrella* is tentative and should merely underline the close relationship with *O. granulosa*.

The material described herein differs from previously recorded fossils assigned to *Ophiomitrella* in the higher number of spine articulations and the finer, less pronounced vertical striation on the outer surface of the lateral arm plates (*Thuy, 2013*). Assignment to a Recent species is precluded by the stratigraphical age of the fossils, implying an implausibly long range; we therefore assign the material described herein to a new species. Notwithstanding the above-mentioned uncertainties in the definition of the genus, *O. floorae* n. sp. fills an important stratigraphic gap between the previously published records of *Ophiomitrella* from the Middle Jurassic (*Thuy, 2013*) and the morphologically similar living species of the genus. Recent members of *Ophiomitrella*, and of the family Ophiacanthidae in general, predominantly live at deep sublittoral to bathyal depths, i.e., between 150 and 2,000 m, (*O'Hara & Stöhr, 2006*; *O'Hara et al., 2017*). Thus, the discovery of *Ophiomitrella floorae* n. sp. aligns with the co-occurring ophiomusaid brittle stars and bourgueticrinine sea lilies in belonging to groups once common and widespread at shallow depths but nowadays restricted to deeper waters (e.g., *Thuy et al., 2012*). Their occurrence at middle sublittoral paleo-depths during the late Maastrichtian is a relict of their mid-Mesozoic occurrence in shallow waters (*Thuy et al., 2012*; *Thuy & Meyer, 2013*; *Thuy, 2013*).

*Ophiomitrella floorae* is one of the first fossil ophiuroids shown to be associated with stalked crinoids. The exceptional discovery of an articulated individual partially wrapped around the stalk of a bourgueticrinine (*Jagt, 2000a*) (Fig. 6) could imply that the ophiuroid lived on the crinoid, with the mouth facing the stalk. The small size and general morphology of *Ophiomitrella floorae* n. sp. conforms to an epizoic lifestyle as commonly observed in living congeners (e.g., *O'Hara & Stöhr, 2006*). However, since the specimen is only partially wrapped around the stalk, we cannot entirely rule out the possibility that the ophiuroid and the crinoid were washed together, with the crinoid accidentally "grabbing" the crinoid stalk before burial. The only unambiguous example of an ophiuroid-crinoid association in the fossil record is the Paleozoic stem ophiuroid *Onychaster* that lived epizoically on stalked crinoids (*Hotchkiss & Glass, 2012*), often found tightly wrapped around the stem or cup of the associated crinoids Living ophiuroids epizoic on rod-shaped hosts like corals generally extend several arms into the water column for feeding and are only tightly coiled around their host when resting (e.g., *Steward & Mladenov, 1994*). Also, ophiuroids dislodged from their substrate, e.g., by strong currents, tend to coil their arms around the disc (*Emson & Wilkie, 1982*) for more rapid sinking. Thus, it seems likely that *O. floorae* n. sp. lived epizoic on bourgueticrinine sea lilies, although we cannot rule out an accidental intertwining of the arms as a result of obrution and/or an attempt of the ophiuroid to free itself after having been smothered.

## CONCLUSIONS

The case of *Ophiomitrella floorae* n. sp. demonstrates that a significant portion of ophiuroid palaeo-biodiversity is easily overlooked. Due to the small size and delicate skeleton of the species, it was much less likely to be noticed than the larger, sturdy-plated and therefore

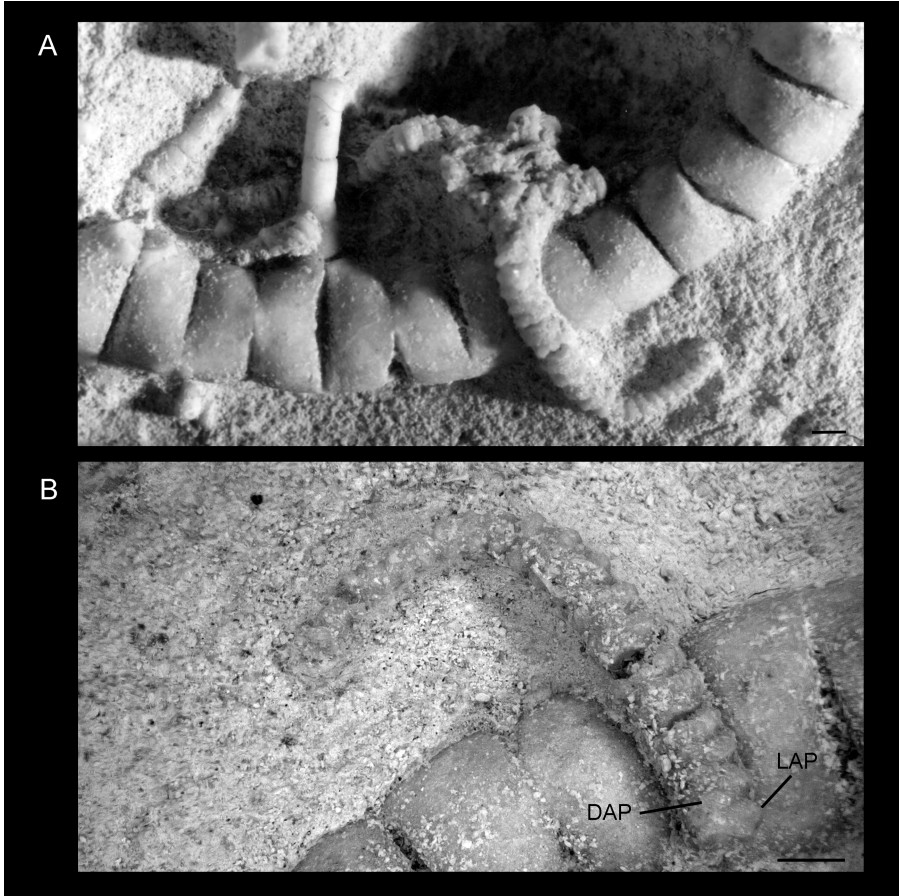

**Figure 6** *Ophiomitrella floorae.* **n. sp., from the lower Gronsveld Member (MaaSt.richt Formation; St. Pieter and ENCI horizons) at the former ENCI-HeidelbergCement Group quarry, MaaSt.richt, the Netherlands. Paratype (NHMM K 3387).** (A) articulated skeleton before removal of the disc and proximal arm segments for scanning electron microscopy; (B) detail of an arm in dorsal view. Abbreviations: DAP, dorsal arm plate; LAP, lateral arm plate. Scale bars equal 1 mm.

more conspicuous co-occurring ophiomusaid and ophiodermatid brittle stars. The single articulated individual was previously too poorly preserved to allow for unambiguous species-level identification (*Jagt, 2000a*). It was only thanks to co-occurring dissociated lateral arm plates that the species could be described, thus underscoring the importance of microfossils in assessing the paleo-biodiversity of taxa with multi-element skeletons such as brittle stars.

# ACKNOWLEDGEMENTS

One of us (JWMJ) thanks the management of the former ENCI-Heidelberg Cement Group quarry (Maastricht) for permission to do fieldwork at their grounds over recent decades, and fellow palaeontologists Mart Deckers, Rudi Dortangs and Marcel Kuypers for support during fieldwork and for donation of echinoderm material. We thank Mart Deckers for providing the photograph used in Fig. 2, and Sabine Stöhr for providing the

scanning electron images used in Figs. 5F–5G. We furthermore thank the reviewers whose comments improved an earlier version of this manuscript.

### Funding
The authors received no funding for this work.

### Competing Interests
The authors declare there are no competing interests.

### Author Contributions
- Ben Thuy, Lea Numberger-Thuy and John W.M. Jagt conceived and designed the experiments, performed the experiments, analyzed the data, prepared figures and/or tables, authored or reviewed drafts of the paper, and approved the final draft.

### Data Availability
Data is available at Figshare: Thuy, Ben; D. Numberger-Thuy, Lea; Jagt, John W.M. (2020): Raw scans for "A new ophiacanthid brittle star (Echinodermata, Ophiuroidea) from sublittoral crinoid and seagrass communities of late Maastrichtian age in the southeast Netherlands". figshare. Figure. https://doi.org/10.6084/m9.figshare.12057933.v1.

Specimens described and illustrated in this article are stored in publicly accessible museum collections: NHMM, Natuurhistorisch Museum Maastricht, Maastricht, the Netherlands (specimens NHMM JJ 5104 and K 3387); SMNH, Swedish Museum of Natural History, Stockholm, Sweden (specimen SMNH-121224); MNHN, Muséum national d'Histoire naturelle, Paris, France (MNHNEcOs22429).

### New Species Registration
The following information was supplied regarding the registration of a newly described species:

Publication LSID: urn:lsid:zoobank.org:pub:9BE69BFD-69FE-4671-BC94-0DF402806A75

Ophiomitrella floorae LSID: urn:lsid:zoobank.org:act:083AAA5B-5C6B-4D4C-99E4-06E89A9F56F7

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
