# Peer review of "A new ophiacanthid brittle star (Echinodermata, Ophiuroidea) from sublittoral crinoid and seagrass communities of late Maastrichtian age in the southeast Netherlands"

_PeerJ, doi:10.7717/peerj.9671_

## Round 0.1 · original submission · Major Revisions

Dear authors,

Given that the three reviewers have not come to a common decision, I have accepted the 'major revisions' suggestion from reviewer two. Please give careful weight to their comments, as well as those made by the other two reviewers,

I look forward to receiving your revised manuscript.

Reviewer 1 ·

Basic reporting

no comments here, see my attachment

Experimental design

no comments here, see my attachment

Validity of the findings

no comments here, see my attachment

Additional comments

no comments here, see my attachment

Annotated reviews are not available for download in order to protect the identity of reviewers who chose to remain anonymous.

·

Basic reporting

Good English throughout in general. One or two corrections are made in the attached pdf.

Introduction could be expanded a little with greater explanation of why now. See general comments for more detail.

Structure and plates in general OK but the conclusions need to be revised as contain much discussion. There is inadequate discussion of the materials in the M&M section.

Experimental design

Research is original.

Questions well defined and meaningful

Validity of the findings

Conclusions are in the main good. I however think that significantly broader discussion is required around the proposed epizoic nature of new species.

Additional comments

Review comments
This paper erects a new species to accommodate material previously assigned to an inappropriate existing taxon, and thereby highlighting important diversity. Whilst much of this work has already be hinted at, or noted in various other publications, the current work is important as it formerly pulls together these notes and thoroughly addresses the implications. I would like to congratulate the authors for this fascinating paper and let them know that I greatly enjoyed reading it.
That said, this paper needs to be revised and restructured in many places. I have made numerous suggestions within the attached PDF where I feel changes or greater elaboration is needed to help the reader more clearly understand what is being attempted. Below I have listed some of the more important points which are based on comments made within the text of the attached pdf. Of the these, the most important to address are; that most of the “Conclusions” are, in fact, mostly discussion points; the introduction needs to be expanded to better contextualise the study, that there is content in the “History of Study” section that is better placed in the material and methods and; that they need better evidence to support their interpretation of this ophiuroid as an epizoic organism.
As such, I am pleased to recommend this paper for publication but only after major revision.

1. The Conclusions are, in the main, not conclusions but a general and very useful discussion about the broader relevance of the findings of this work. As such, the paper needs to be re-structured accordingly.
2. Please expand the Introduction slightly to say specifically which aspects of these specimens are providing the impetus for the study. Something along the lines of it has not been adequately described/accurately assigned and this has broader implications but it would help to state this explicitly and why it is being done now.
3. I think the evidence for O. floorae being epizoic on Bourgueticrinids is poor. This association is as likely, if not more likely, to be the result of both organisms being washed together during the obrution event that formed the associated Lagerstatten from which these fossils were collected from. This is because only one arm is draped around the stem, not 4, or all 5 arms closely clinging to the stem or cup, as seen in multiple examples from the Carboniferous. That they lived in the same environment, there is no doubt, but I do not think the current evidence of “general resemblance” to known modern epizoic ophiocanthids is strong enough to interpret the fossil co-occurrence as an epizoic relationship. Please also, see further comments made on lines 224-225. As such, significantly more evidence of epizoic behaviour in O. floorae is needed than is currently provided or the various statements regarding its epizoic habit need to be revised throughout the paper (in the “Conclusions”, Abstract and, Introduction).
4. Whilst I can see that images of the association between the crinoid and ophiuroid have been published previously (Jagt 2000a) an image of it must be included in this paper if the authors want to maintain that the relationship is epizoic or discuss the palaeoecological implications. As such, in Figure 5 (or a separate figure) please show the association between the ophiuroid and the bourgueticrinid stem.
5. In the log (figure 3), I am assuming that the black globular shapes are a representations of flint bands. There are also other shapes on here which may indicate bioturbation. Please can a key be included? The log would be further improved if chronological dates are added (these are mentioned in the text so it should not be too hard to do) and that the columns are labelled (particularly for the column with IVa-1 etc. as I am unsure specifically what these refer to). Finally, can they please show where the specimen was located within the log. In the text it is fairly explicit as to where it is from but the arrow on the log instead points to “one of the more spectacular storm levels”. Whilst interesting, storm levels have little apparent relevance to the rest of the paper.
6. I am confused about the status of the other specimens assigned to Ophiocantha? danica by Jagt 2000a but not mentioned at all in the current paper. I assume that they are not part of the new species but I think it would be valuable to add some discussion about the status of it as there is so little material currently included in the new species and yet there is published reference to more material from the same site that was formerly placed within the same taxa by one of the co-authors.
7. See pdf for further comments.

·

Basic reporting

The English is excellent.

Experimental design

No experimental design per se in a paleontological taxonomy study. The methods and examinations follow standard protocol for this type of study.

Validity of the findings

I agree that the fossil is an ophiacanthid and due to the fact that it was epizoic on a crinoid, it is more likely to be an Ophiomitrella than an Ophiacantha. However, it would be helpful for readers if you would explain why you exclude other possible genera. The LAP is very similar not only to Ophiomitrella but also to Ophiacantha. Without knowing the radial shield and possible granules on the disc, why can't it be an Ophiacantha?

You state (row 199) that Ophiomitrella is paraphyletic, which would mean that all Ophiomitrella species share a common ancestor but one or more groups that also share this ancestor are excluded. On the phylogenies by O'Hara (latest version in Christodoulou et al. 2019), Ophiomitrella includes species with different ancestors, which means it is polyphyletic (as is stated in O'Hara et al. 2018 not 2017). As long as we don't know in which group the type species falls, we cannot decide, which of these are to be considered Ophiomitrella, but there are 3 clades. Can you really decide that floorae is closest to the clade that contains clavigera and conferta? Why not the clade with globulifera or the one with stellata? Perhaps you can comment a little more on this. Considering that Ophiacantha is likewise polyphyletic and intermingled with Ophiomitrella on the tree, this is difficult and should be discussed a little more broadly.

Additional comments

Another paleontological mystery has been cleared up, very nice.
Just out of curiosity, I do not have the original paper (Jagt 200) at hand and wonder how this animal looked when it was wrapped around the crinoid stalk. Was it removed or are we suppose to see this in the images? The ophiuroids wrapped around crinoid arms that I have seen were orientated with the mouth towards the crinoid. Here it was only one arm that attached to the stalk, but was it broken off or how?

---

## Round 0.2 · Minor Revisions

Dear authors,

Based on the comments from the reviewers I have accepted the decision of 'minor revisions'.

As you will see from the comments, the changes to be made are minor and will be easy for you to make.

I look forward to receiving your revised manuscript.

Reviewer 1 ·

Basic reporting

no comment

Experimental design

no comment

Validity of the findings

no comment

Additional comments

I have already wrote before that this is very interestging MS that should be published in PeerJ. I see that in its current form, all corrections and suggestions proposed by me and other reviewers have been made. Basically I would suggest publishing the article as it is, but I have found some editorial mistakes that need to be corrected (Am attaching pdf).

Annotated reviews are not available for download in order to protect the identity of reviewers who chose to remain anonymous.

·

Basic reporting

All fine

Experimental design

Fine

Validity of the findings

Fine

Additional comments

Whilst, I remain unconvinced of the geological evidence for an epizoic relationship between this individual ophiuroid and the crinoid, I can see that it resembles some modern epizoic ophiuroids. The amendments made in the paper addressing this point are fine. As far as I am concerned, all the other changes suggested by the reviewers have been thoroughly addressed and it is a pleasure to recommend publication. My congratulations to the authors for such a nice paper.

·

Basic reporting

The English is excellent

Experimental design

There is no experiment as such but the methods are appropriate

Validity of the findings

Valuable study.

Additional comments

The authors have revised the manuscript diligently and I have only a few minor corrections to suggest.

rows 51-52: 'attempt at', I think, the verb attempt does not go with 'at'
row 111: change danica to small d
row 225: the historical samples are not rare (but very valuable), the species seems to be rare in collections
Fig 5F-G seems to be the previous caption in the pdf but is changed in the Word document, and please attribute these to S. Stöhr.

---

## Round 0.3 · accepted · Accept

Dear authors,

Thank you for your revised manuscript. I can confirm that it has been accepted for publication in PeerJ.

You will be contacted shortly by the production staff regarding the proof stages.

Thank you for choosing PeerJ as your publishing venue, and I hope you will use us again in the future.